# Metformin Inhibits Tumor Metastasis through Suppressing Hsp90α Secretion in an AMPKα1-PKCγ Dependent Manner

**DOI:** 10.3390/cells9010144

**Published:** 2020-01-07

**Authors:** Yuanchao Gong, Caihong Wang, Yi Jiang, Shaosen Zhang, Shi Feng, Yan Fu, Yongzhang Luo

**Affiliations:** 1The National Engineering Laboratory for Anti-Tumor Protein Therapeutics, Tsinghua University, Beijing 100084, China; gongyc14@mails.tsinghua.edu.cn (Y.G.); wangch15@mails.tsinghua.edu.cn (C.W.); jiang-y17@mails.tsinghua.edu.cn (Y.J.); zhangss14@mails.tsinghua.edu.cn (S.Z.); fengs14@mails.tsinghua.edu.cn (S.F.); fuyan@tsinghua.edu.cn (Y.F.); 2Beijing Key Laboratory for Protein Therapeutics, Tsinghua University, Beijing 100084, China; 3Cancer Biology Laboratory, School of Life Sciences, Tsinghua University, Beijing 100084, China

**Keywords:** metformin, Hsp90α, metastasis, AMPKα1, PKCγ

## Abstract

Metformin has been documented in epidemiological studies to mitigate tumor progression. Previous reports show that metformin inhibits tumor migration in several cell lines, such as MCF-7 and H1299, but the mechanisms whereby metformin exerts its inhibitory effects on tumor metastasis remain largely unknown. The secreted proteins in cancer cell-derived secretome have been reported to play important roles in tumor metastasis, but whether metformin has an effect on tumor secretome remains unclear. Here we show that metformin inhibits tumor metastasis by suppressing Hsp90α (heat shock protein 90α) secretion. Mass spectrometry (MS) analysis and functional validation identify that eHsp90α (extracellular Hsp90α) is one of the most important secreted proteins for metformin to inhibit tumor cells migration, invasion and metastasis both in vitro and in vivo. Moreover, we find that metformin inhibits Hsp90α secretion in an AMPKα1 dependent manner. Our data elucidate that AMPKα1 (AMP-activated protein kinase α1) decreases the phosphorylation level of Hsp90α by inhibiting the kinase activity of PKCγ (protein kinase Cγ), which suppresses the membrane translocation and secretion of Hsp90α. Collectively, our results illuminate that metformin inhibits tumor metastasis by suppressing Hsp90α secretion in an AMPKα1 dependent manner.

## 1. Introduction

Metformin is a widely prescribed biguanide derivative used as the first-line therapy for type 2 diabetes. Epidemiological studies have shown that metformin reduces the incidence of cancer and cancer-related mortality in diabetic patients [1]. Since then, metformin has been attracting interests due to its anti-tumor effects. Some exciting reports show that metformin inhibits tumor progression alone or in combination with other drugs [2,3,4,5]. Varied mechanisms underlying the anti-tumor effect of metformin have been demonstrated in a number of studies [6,7,8,9,10,11,12,13,14]. Metformin can cause demethylation of DNA and lead to up-regulation of some encoding genes and non-coding RNAs, such as miR-192-5p, miR-584-3p, miR-1246, EGF containing fibulin-like extracellular matrix protein 1 (EFEMP1) gene and secretory carrier membrane proteins (SCAMP3) gene [6]. Many studies report that metformin can suppress the growth of tumor by leading to apoptosis and autophagy [7,8,9]. Several studies also show that metformin can inhibit the epithelial to mesenchymal transition (EMT) process [10,11,12] and regulate the immune system to suppress tumor progression [13,14].

As for the downstream regulators of metformin, AMPK may be the most important one. The AMP-activated protein kinase (AMPK) acts as a physiological cellular energy sensor and plays a central role in regulating cellular metabolism to maintain energy homeostasis [15]. AMPK is a heterotrimeric complex consisting of a catalytic α-subunit and regulatory β- and γ-subunits. The phosphorylation of AMPKα at threonine 172 mainly by liver kinase B1 (LKB1) is required for AMPK activation [16]. Many studies show that AMPK largely functions to suppress tumorigenesis via controlling cell growth, metabolism and polarity [17,18,19]. In supporting these observations, AMPK activation by small molecules has been shown to suppress tumor progression in several tumor models [20,21].

The combination of metformin and antibodies targeting extracellular proteins or receptors has achieved good results in some tumor research [22,23] or is being evaluated in clinical studies [24]. The tumor secretome is a rich reservoir for cancer drug targets [25,26] and understanding the effect of metformin on tumor secretome is well worth exploring. Extracellular Hsp90α in tumor secretome is such a protein that highly associates with tumor progression [27]. The 90-kDa heat shock protein (Hsp90) is an essential and abundant intracellular molecular chaperone that assists in various biological processes and stabilizes hundreds of client proteins [28,29]. Hsp90 has two isoforms: Hsp90α and Hsp90β [30]. Interestingly, Hsp90α has been found to be localized on the cell membrane and secreted into the extracellular space, also called eHsp90α (extracellular Hsp90α) [31,32]. Previous studies prove that plasma eHsp90α is a good broad-spectrum biomarker for cancer [33,34]. Furthermore, China Food and Drug Administration (CFDA) has approved the quantitative enzyme-linked immunosorbent assay (ELISA) kit of plasma eHsp90α to be used in lung or liver cancer detection, including disease status assessment and outcome prediction.

The extracellular Hsp90α (eHsp90α) has been found to play important roles in tumor cells migration, invasion and metastasis in various cancer types [35,36,37]. eHsp90α can activate pro-migratory signaling pathways and EMT process by interacting with low-density lipoprotein receptor-related protein (LRP-1) [38,39]. In addition, eHsp90α can also stabilize and activate matrix metalloproteinase-2 (MMP-2) to degrade extracellular matrix (ECM), which is essential for tumor cells to invade out from the primary tumor and form metastases [32,40]. Based on the above research, cell impermeable inhibitors and neutralization antibodies targeting eHsp90α have been shown to inhibit tumor metastasis significantly [41,42], which means eHsp90α is a potential tumor drug target in cell secretome.

In this study, we find metformin exerts its anti-tumor effects through suppressing Hsp90α secretion in an AMPKα1 dependent manner. AMPKα1 activation inhibits the kinase activity of PKCγ, which then results in the decreased phosphorylation level of Hsp90α and the suppression of the membrane translocation and secretion of Hsp90α.

## 2. Materials and Methods

### 2.1. Cell Culture, Reagents and Antibodies

Human lung cancer cell lines H1299, A549, human breast cancer cell lines MCF-7, MDA-MB-231 were obtained from the American Type Culture Collection (Manassas, VA, USA) and maintained in DMEM supplemented with 10% FBS and 1% antibiotic. Metformin (ab120847), AICAR (ab120358) and Compound C (ab120843) were purchased from Abcam (Shanghai, China). Antibodies against AMPKα1 (2795S), AMPKα2 (2757S), AMPKα (2532S), Phospho-AMPKα (Thr172) (2531S), PKCγ (43806S), Na, K-ATPase (3010S) were purchased from Cell Signaling Technology (Danvers, MA, USA). Antibody against Phospho-PKCγ (Thr514) (AF8347) was purchased from Affinity (Cincinnati, OH, USA). Antibodies against Phospho-serine/threonine (ab15556) and β-actin (ab8227) were purchased from Abcam. Antibody against Hsp90α and recombinant Hsp90α were provided by Protgen (Beijing, China). Detailed information about the antibodies is shown in Appendix A.

### 2.2. Quantitative RT-PCR

The TRIZOL (Invitrogen, Thermo Fisher Scientific, Wilmington, DE, USA) reagent was used to isolate the total RNA form tumor cells. Then the First Strand cDNA Synthesis Kit (Thermo Fisher Scientific) was used for the synthesis of cDNA and the expression of relative genes was measured by the Mx300 system. The expression of β-actin was used as the control for relative quantitation of the genes. Every experiment was conducted three times independently.

### 2.3. Western Blot

Cell lysate sample and condition medium were collected from cells cultured in plates and subjected to SDS-PAGE and transferred to polyvinylidene difluoride membrane (Millipore, Billerica, MA, USA). The membrane was blocked with 5% fat-free milk and incubated with specific primary antibodies at 4 °C overnights. Then the membrane was incubated with HRP-conjugated secondary antibodies. The expression of proteins was detected with an enhanced chemiluminescence system (Thermo Fisher Scientific).

### 2.4. Lentivirus Infection

Lentivirus infection was performed to construct AMPKα1 and PKCγ-overexpressing MCF-7 and H1299 cell lines. Three plasmid system was used for the transfection. The plasmid system included pVSVG (the envelope plasmid), psPAX2 (the packaging vector) and the pLentiCMV plasmid containing AMPKα1 or PKCγ-coding sequence the three plasmids were co-transfected into HEK 293T and the conditioned medium (CM) was collected after 72 h. Then, the conditioned medium was added into H1299 or MCF-7 cells after being filtered by an 0.45 μm filter. After 72 h, fresh culture medium was used to replace the medium from H1299 or MCF-7 cells. After that, Blasticidin (Selleck, Houston, TX, USA) was used for the stably transfected cells selection. The PCR primer sequences for the cloning of AMPKα1 and PKCγ were shown in Appendix A. The AMPKα1 knockdown cell line was constructed following the same protocol using shRNA.

### 2.5. siRNA Interference

Down-regulation of AMPKα1, AMPKα2 and PKCγ was obtained by siRNAs transfection. The siRNAs were purchased from GenePharma. The sequence information was shown in Appendix A. Following the manufacturer’s protocol, siRNAs were transfected into tumor cells by using Lipofectamine 200 reagent (Invitrogen). After 48 h transfection, Western blot was applied to evaluate the knockdown efficiency.

### 2.6. Cell Proliferation Assay

1 × 10^4^ cells were seeded into a cell of a 96-well plate. After 24 h or 48 h, the conditioned medium was removed and 100 μL serum-free medium with 10 μL CCK-8 solution was added. Then, the cells were cultured for about 30 min at 37 °C. The proliferation ability of tumor cells was measured by determining the absorbance at 450 nm. Every experiment was conducted three times independently.

### 2.7. Plasma Membrane Extraction

Cell membrane was separated by using Plasma Membrane Protein Isolation Kit (Invent Biotechnologies, Plymouth, MN, USA). According to the protocol, the cells were collected and transferred into protein extraction filter cartridges. Then the mixture was centrifuged at 14,000× *g* for 30 s, and the pellet was resuspended. After centrifugation at 3000× *g* for 1 min, the supernatant was again transferred. After that, the mixture was centrifuged at 16,000× *g* for 30 min, and the pellet was saved (plasma membrane).

### 2.8. Cell Invasion Assay and Cell Migration Assay

The ability of tumor cells invasion was measured by using transwell system with Matrigel coated inserts. Briefly, tumor cells were seeded in the upper chamber of 8 μm Millicell coated with Matrigel. Reagents including metformin, Hsp90α antibody, recombinant Hsp90α protein or IgG were added to the lower chamber with 1% FBS medium. Then we counted the migrated cells in eight fields per cell randomly by using optical microscope at 40× magnification. After that, we measured the cells relative invasion ability by normalizing the number of migrated cells to the control groups. The steps in the cell migration assay were similar to the cell invasion assay, and the only difference being that the Millicell used in the cell migration assay were not coated with Matrigel.

### 2.9. Co-Immunoprecipitation Assay (Co-IP)

Tumor cells were suspended with cold PBS and then centrifuged at 3000 rpm for about 5 min. The cell pellet was lysed by using lysis buffer at 4 °C for 20 min. After that, the mixture was centrifuged at 14000 rpm for 10 min and the supernatant was collected. Then the indicated antibodies and protein A Sepharose beads were incubated with supernatant for at least 12 h at 4 °C. We prepared western blot protein samples by boiling beads with the sample buffer (1% SDS, 1 mM dithiothreitol) at 100 °C. The lysis buffer contained 150 mM NaCl, 20 mM Tris, 0.5% NP40 and phosphatase and protease inhibitors.

### 2.10. Mass Spectrometry

The whole gel slices containing protein bands were excised and digested by sequencing grade modified trypsin following the SDS-Page. After that, liquid chromatography mass spectrometry was used to analyze these peptides and we used the Swiss Prot database to do the piloting. Label-free quantification of the MS data was performed in the MaxQuant environment.

### 2.11. Flow Cytometry Analysis

Cells were collected by using cold PBS and primary antibodies were added into and incubated with the mixture for 1 h on ice. After washed with cold PBS, the fluorescein conjugated secondary antibodies were added into and incubated with the mixture for 30 min on ice. After washing with cold PBS twice, a FACSAria III system (BD Biosciences, San Jose, CA, USA) was used to analyze the cells.

### 2.12. Exosomes Isolation

Exosomes were isolated by using the miRCURY Exosome Cell Kit following the manufacturer’s instructions (Qiagen, Benelux B.V., Germany). Ten mL conditioned medium was mixed with Precipitation Buffer B, and vortexed thoroughly and then incubated for 60 min at 2–8 °C. After that, the mix was centrifuged at 3200× *g* for 30 min at 20 °C. The supernatant was removed and discarded. The pellet was resuspended by using 100 μL resuspension buffer for exosome analysis.

### 2.13. Animal Experiments

The Institutional Animal Care and Use Committees of Tsinghua University approved the animal studies and the approved number is 16-LYZ4. For the orthotopic breast tumor implantation assays, two groups of MCF-7 cells (10^7^ cells in 100 μL of PBS containing 50 μL Matrigel) (Corning, New York, NY, USA) were injected into the fat pad of 6-week-old mice. After 10 days, one group of mice was treated with saline and the other group was treated with metformin (200 mg/kg of body weight once a day) orally.

For the orthotopic lung tumor implantation assays, four groups of H1299 cells (2 × 10^6^ cells) were injected into the left pulmonary lobes of nude mice of 6-week-old mice. The H1299 cells used for orthotopic tumor implantation experiments were stably labeled with a luciferase expressing vector and were monitored by weekly bioluminescent imaging. After 10 days, the first group of mice was treated with saline, the second group was treated with recombinant Hsp90α protein, the third group was treated with metformin, and the fourth group was treated with both recombinant Hsp90α protein and metformin. Metformin was orally administered 200 mg/kg of body weight once per day. Recombinant Hsp90α protein was injected with 10 mg/kg per mice twice per week via tail vein. 40 days after implantation, all the mice were killed by carbon dioxide anaesthesia and the lungs and livers were removed and fixed in 10% formalin. It has been shown that the concentration of metformin had an important effect on its action [43]. The concentration used in this study was based on the previous research [6].

### 2.14. Statistical Methods

The data was represented as means ± standard deviations (SDs) or means ± standard errors of the mean (SEMs). Statistical analysis was performed using the two-tailed, unpaired Student’s t-tests with the GraphPad Prism (GraphPad Software, San Diego, CA, USA). *p* values < 0.05 were considered as significant difference.

## 3. Results

### 3.1. Metformin Inhibits Hsp90α Secretion both In Vitro and In Vivo

Four cell lines, H1299, A549, MCF-7 and MDA-MB-231, were treated with metformin (Met) and the cell migration was examined using the trans-well system. The cell migration ability was significantly suppressed in H1299 and MCF-7 cell lines (Figure 1A,B) but not A549 and MDA-MB-231 cell lines (Appendix A) after metformin treatment. These results showed the suppressive effects of metformin on tumor migration differed in different cell lines. Metformin had no effect on tumor proliferation at 48 h (Figure 1C and Appendix A) and 24 h (Appendix A). H1299 and MCF-7 cell lines were chosen for the following experiments. To explore whether metformin had an effect on tumor secretome, conditioned medium (CM) derived from H1299 and MCF-7 cells before and after metformin treatment was collected. Following SDS-PAGE (Appendix A), mass spectrometry was conducted to analyze the differences in the conditioned medium. As depicted in Figure 1D, 91 proteins with a score over 100 were consistently identified in both H1299 and MCF-7 cell lines. Among these proteins, six proteins’ concentration in conditioned medium changed more than twofold after metformin treatment (<2.0) as measured by MS (Table 1). Hsp90α was chosen and hypothesized to modulate the function of metformin for several reasons: (1) the main function of eHsp90α was promoting tumor cells migration, invasion and metastasis, but not tumor cells proliferation. (2) the change of eHsp90α after metformin treatment was relatively large (Ctrl/Met was 3.24). (3) the level of eHsp90α was correlated with the metastasis of multiple cancer types. eHsp90α was detected in concentrated conditioned medium and the results showed that the amount of eHsp90α was decreased after treated with metformin (Figure 1E,F). Metformin had no effect on Hsp90α expression (Figure 1G). In order to confirm this phenomenon in vivo, we orthotopically injected H1299 cells with luciferase activity into the left pulmonary lobes of nude mice to generate primary tumors and plasma was collected for the detection of eHsp90α using ELISA assay. We obtained the results that metformin treated group had a lower eHsp90α level than the control group (Figure 1I) while the tumor burden was not significantly different (Figure 1H and Appendix A). We also injected MCF-7 cells into the fat pad of nude mice and then detected the plasma eHsp90α. The results also showed that metformin decreased the level of eHsp90α in plasma (Figure 1J) and there was no significant difference in the tumor weight (Appendix A). Thus, these results demonstrate that the secretion of Hsp90α is inhibited by metformin both in vitro and in vivo.

### 3.2. Metformin Inhibits Tumor Metastasis through Suppressing Hsp90α Secretion

To explore the contribution of eHsp90α to the inhibitory effects of metformin on tumor metastasis, tumor cells were treated with recombinant Hsp90α (rHsp90α) and the migration and invasion ability were detected by using trans-well assay. The data showed that rHsp90α could rescue the inhibitory effect of metformin on migration (Figure 2A,C and Appendix A) and invasion (Figure 2B,D and Appendix A). To further validate these findings in mice model, H1299 cells with luciferase activity were injected orthotopically into the nude mice. After one month, luciferase activity was monitored by bioluminescent imaging. It was shown that metformin suppressed spontaneous tumor metastasis significantly and rHsp90α rescued the inhibitory effect (Figure 2E,F). At the same time, the fluorescence intensity of primary tumor was no significantly different (Figure 2E and Appendix A). And also, metformin treatment had no effect on the mice weight (Appendix A). Then the lung was removed for H&E staining to detect tumor cells local invasion.

As shown in Figure 2G,H, metformin inhibited tumor cells local invasion significantly while rHsp90α abolished this kind of inhibitory effect. The liver was also eviscerated for the detection of tumor metastasis (Figure 2I), and we obtained the similar results (Figure 2J,K). The results above show that metformin inhibits tumor metastasis by suppressing Hsp90α secretion.

### 3.3. AMPKα1 but not AMPKα2 Mediates Hsp90α Secretion

Next, we sought to determine how metformin regulated Hsp90α secretion. Early studies report that AMPK is one of the important downstream regulators of metformin [15], but its function in Hsp90α secretion has never been reported. Tumor cells were treated with AMPK activator (AICAR) and inhibitor (Compound C) and the secretion of Hsp90α was detected. We found that AICAR treatment inhibited Hsp90α secretion (Figure 3A) while Compound C treatment facilitated Hsp90α secretion (Figure 3B), which demonstrated that AMPK activation could inhibit Hsp90α secretion. AMPK is a kinase consisting of three subunits: AMPKα, AMPKβ and AMPKγ, among which AMPKα has the kinase activity [15]. AMPKα has two isoforms: AMPKα1 and AMPKα2. Several studies have shown that the two isoforms had distinct functions [44,45]. We sought to determine which isoform had the inhibitory effect on Hsp90α secretion. We used siRNAs to knock down AMPKα1 and AMPKα2 separately in tumor cells. The efficiency of siRNAs was measured by Western blot (Appendix A). The #3 siRNA for AMPKα1 and #2 siRNA for AMPKα2 were chosen for the knockdown experiments. Notably, AMPKα1 knockdown facilitated Hsp90α secretion significantly while AMPKα2 knockdown had no impact on the secretion of Hsp90α (Figure 3C,D). Both AMPKα1 and AMPKα2 knockdown had no effect on the expression of Hsp90α in cells (Figure 3E and Appendix A). The results above illustrate that AMPKα1 and AMPKα2 play different roles in regulating the secretion of Hsp90α and AMPKα1 is the main regulator.

### 3.4. Metformin Inhibits Hsp90α Secretion in an AMPKα1 Dependent Manner

To elucidate whether the inhibitory effect of metformin on Hsp90α secretion was dependent on AMPKα1, AMPKα1 knockdown (AMPKα1 KD) and overexpression (AMPKα1 OV) cell lines were constructed by stable transfection with lentivirus based on H1299 and MCF-7 cells. Then AMPKα1 WT, AMPKα1 KD and AMPK OV cells were treated with metformin, respectively.

We found that the inhibitory effect of metformin on Hsp90α secretion was diminished in AMPKα1 KD cells (Figure 4A and Appendix A) while enhanced in AMPKα1 OV cells. (Figure 4B and Appendix A). Based on these observations, we demonstrated that metformin inhibited Hsp90α secretion through activating AMPKα1. We also confirmed these results in the migration and invasion assays. Hsp90α antibody abolished the promoting effects of AMPKα1 knockdown on cell migration and invasion (Figure 4C–F). Simultaneously, rHsp90α rescued the inhibitory effects of AMPKα1 overexpression (Figure 4G–J). The similar results were also shown in MCF-7 cells (Appendix A) and the quantified results were shown in Figure 4K–N. Collectively, the data above demonstrate that metformin inhibits Hsp90α secretion in an AMPKα1 dependent manner.

### 3.5. AMPKα1 Decreases the Phosphorylation Level of Hsp90α and Suppresses Hsp90α Membrane Translocation

Hsp90α has been reported to be secreted in exosomes due to lack of a signaling peptide [36,38]. So, we wondered whether AMPKα1 had an effect on exosomes secretion. Exosomes Quantitative Kit was used to measure exosomes in conditioned medium. The results showed that the amount of exosomes was not significantly different among AMPKα1 WT, KD and OV tumor cells (Figure 5A,B). CD63 and CD9, two multivesicular body (MVB) markers, were also detected. The amount of these markers in conditioned medium was also the same among the three kinds of cells (Figure 5C). These data demonstrate that AMPKα1 has no effect on exosomes secretion.

Phosphorylation has been reported to be associated with the membrane translocation and secretion of Hsp90α [46,47,48]. Then we hypothesized that AMPKα1 might have an impact on the Hsp90α phosphorylation. To prove this, Hsp90α was pulled down by a specific antibody and the Ser/Thr phosphorylation level was detected. We found that AMPKα1 knockdown increased the phosphorylation level (Figure 5D) while AMPKα1 overexpression decreased the phosphorylation level of Hsp90α (Figure 5E), which implied that AMPKα1 inhibited Hsp90α secretion by decreasing the phosphorylation level of Hsp90α.

It has been reported that the membrane translocation of Hsp90α was highly associated with its secretion [49]. Cell membrane was extracted and the Hsp90α on the cell membrane was measured. The results showed that AMPKα1 knockdown facilitated Hsp90α membrane translocation (Figure 5F) while AMPKα1 overexpression downregulated the Hsp90α membrane translocation in tumor cells (Figure 5G). To further confirm this result, flow cytometry experiments without cell permeabilization were conducted by using AMPKα1-WT, KD, OV cells. We obtained the similar results (Figure 5H and Appendix A). All the results above demonstrate that AMPKα1 inhibits Hsp90α secretion by decreasing the phosphorylation and membrane translocation of Hsp90α.

### 3.6. AMPKα1 Inhibits Hsp90α Phosphorylation, Membrane Translocation and Secretion by Suppressing the Kinase Activity of PKCγ

Hsp90α has been reported to be phosphorylated by several kinases, in which Protein Kinase A (PKA) and Protein Kinase C (PKC) were reported to regulate the membrane translocation or secretion of Hsp90α [46,47]. Previous studies showed that the kinase activity of PKC could be inhibited by AMPK [50] while PKA acted as an upstream kinase of AMPK [51], which meant PKC could be a bridge between AMPK and Hsp90α. PKC has several isoforms [52] and our group has reported that only PKCγ was correlated with the membrane translocation or secretion of Hsp90α [47]. We wondered whether AMPKα1 suppressed Hsp90α secretion through inhibiting the kinase activity of PKCγ. As depicted in Figure 6A,B, the kinase activity of PKCγ was enhanced in AMPKα1 KD cells but inhibited in AMPKα1 OV cells.

It has been shown that the phosphorylation of Hsp90α by PKCγ triggered the release of PKCγ from Hsp90α [48]. Hsp90α was pulled down and the protein level of PKCγ was measured. More PKCγ protein dissociated from Hsp90α in AMPKα1 KD cells compared with AMPKα1 OV cells (Figure 6C and Appendix A). These results show that AMPKα1 inhibits the kinase activity of PKCγ.

To prove whether AMPKα1 decreased the phosphorylation level of Hsp90α through inhibiting PKCγ, PKCγ was knocked down in AMPKα1-KD cells and the phosphorylation level of Hsp90α was measured. The results showed that the increased phosphorylation of Hsp90α resulted by AMPKα1 deficiency was almost abolished (Figure 6D and Appendix A). Moreover, the level of Hsp90α membrane translocation and secretion that were up-regulated in AMPKα1-defective cells were also diminished (Figure 6F and Appendix A). On the other hand, PKCγ was overexpressed in AMPKα1-OV cells. The overexpression of PKCγ abolished the inhibitory effects of AMPKα1 on Hsp90α phosphorylation, membrane translocation and secretion (Figure 6E,G and Appendix A). All the results above point out that AMPKα1 mediates Hsp90α membrane translocation and secretion by suppressing the kinase activity of PKCγ.

## 4. Discussion

Research interest on the anti-tumor effects of metformin was sparked in 2005, when a reduced risk of cancer in type 2 diabetes patients treated with metformin was shown [1]. Since then, studies examining the anticancer mechanisms of metformin have gradually increased. Metformin inhibits mTOR and more pronounced 4E-binding protein (4EBP1) through reducing site-specific phosphorylation [53]. Inhibition of the mTOR/S6/4EBP1 axis is known to impair protein synthesis and cell proliferation [53]. Metformin is also shown to exert regulative functions on miRNAs [6], which results in growth inhibition and cell viability impairment. In addition, metformin has been reported to inhibit epithelial-to-mesenchymal transition (EMT) to suppress tumor progression [10]. Other than the above molecular mechanisms, metformin is also involved in tumor cells autophagy [7,8], apoptosis [9], epigenetic features [6] and immune responses [13,14]. In general, relative few reports focus on the effects of metformin on tumor metastasis and most molecular mechanisms about metformin are involved in intracellular signalling pathways. Here, we study the role of metformin from another perspective that whether metformin has an effect on extracellular proteins. Cancer cell-derived secretome was analyzed by mass spectrometry and eHsp90α was identified as one of the most important proteins affected by metformin. eHsp90α has been reported to be associated with tumor cells migration, invasion and metastasis in various cancer types [27]. In this research, the in vitro and in vivo studies show that metformin suppresses tumor progression by decreasing the level of eHsp90α. Tumor cell is a complex and dynamic system, and the composition of extracellular proteins has been reported to be changed in response to various stress [54]. Our results indicate that as an external stimulus, metformin can suppress tumor metastasis by altering the composition of extracellular proteins. eHsp90α promotes tumor progression via diverse mechanisms [27], suppressing Hsp90α secretion by metformin may demonstrate the diverse utility of metformin in tumor therapy. Still, rHsp90α cannot fully rescue the inhibitory effect of metformin on tumor metastasis, which means there are other unknown pathways for metformin that can inhibit tumor metastasis.

What’s more, previous research has shown different effects of metformin on different cancer subtypes [55], but the underlying mechanisms remain largely unknown. In this research, four cell lines are used to detect the effect of metformin. The migration of H1299 and MCF-7 cell lines are inhibited by metformin (Figure 1A,B) while A549 and MDA-MB-231 cell lines are non-responsive to metformin (Appendix A). Correspondingly, the secretion of Hsp90α is suppressed by metformin in H1299 and MCF-7 cell lines (Figure 1E) but not in A549 and MDA-MB-231 cell lines (Appendix A). These results indicate that the distinct functions of metformin may be due to the different effects on Hsp90α secretion. On one side, plasma eHsp90α detection may be an effective way to identify the patients who are metformin responsive. This combination of drug therapy and biomarker detection will definitely improve the therapeutic efficiency of metformin. On the other side, cell impermeable inhibitors and neutralization antibodies targeting eHsp90α have been reported as drug candidates to suppress tumor metastasis [41,42]. Patients who are not responsive to metformin can be treated with drugs targeting eHsp90α.

As a downstream effector of the tumor suppressor LKB1, AMPK is considered to have an anti-tumor effect. AMPKα activators are being developed as drug candidates for cancer treatment [20,21]. In this research, the experimental data show that AMPKα1 and AMPKα2 play different roles in mediating Hsp90α secretion and tumor metastasis. AMPKα1 knockdown facilitates the secretion of Hsp90α. However, AMPKα2 knockdown has no effect on the Hsp90α secretion. This isoform-specific regulation of Hsp90α secretion raises the possibility that selective modulation of AMPKα1 can delay the progression of cancers.

AMPKα have been reported to interact with Hsp90 directly [56]. The binding of Hsp90 to AMPK stabilizes the three subunits of the AMPK complex, suggesting that Hsp90 may serve as an endogenous positive modulator [56]. But it is not clear whether AMPKα has effects on the secretion of Hsp90α. In spite of the clear importance of Hsp90α in extracellular space, our current understanding of the mechanisms about controlling Hsp90α secretion remains fragmentary. Cell stressors such as heat shock, hypoxia and proteasome inhibitors stimulate Hsp90α translocation to the membrane and/or secretion [27]. Since Hsp90α does not have a signaling peptide, it is secreted through unconventional pathways, such as exosomes, secretory vesicles or some other unknown pathways [36]. In this research, we find the amount of exosomes secreted outside tumor cells is not affected by AMPKα1, but AMPKα1 decreases the phosphorylation level of Hsp90α. Phosphorylation has been reported to be associated with the secretion of various proteins [57,58]. Our group has also shown that the phosphorylation of Hsp90α played an important role in regulating Hsp90α membrane translocation and secretion [46,47,48]. The secretion of Hsp90α is determined by the phosphorylation status at residue Thr-90, regulated by protein kinase A (PKA) and protein phosphatase 5 (PP5) [46]. Moreover, protein kinase C gamma (PKCγ) is involved in the cytosol-to-membrane translocation of Hsp90α [47,48]. Here, our data shows that AMPKα1 activation inhibits Hsp90α membrane translocation and secretion by decreasing the phosphorylation level of Hsp90α through suppressing the kinase activity of PKCγ, which demonstrates that PTM (post translational modification) bridges between metformin and Hsp90α secretion.

## 5. Conclusions

As shown in Figure 7, metformin activates the kinase activity of AMPKα1. The activated AMPKα1 decreases the phosphorylation level of Hsp90α by inhibiting the kinase activity of PKCγ, which suppresses the membrane translocation and secretion of Hsp90α. The decreased level of secreted Hsp90α leads to the decreased ability of tumor cells migration, invasion and metastasis.

## Figures and Tables

**Figure 1 cells-09-00144-f001:**
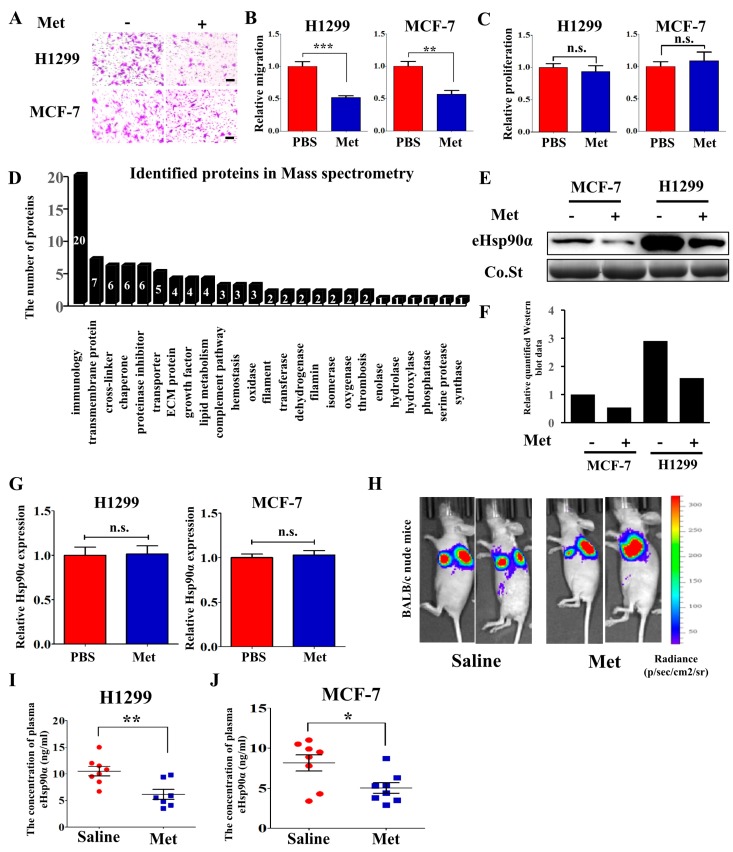
Metformin inhibits Hsp90α secretion both in vitro and in vivo. (**A**) Representative images and (**B**) quantified results of H1299 and MCF-7 cells migration assay treated with PBS or metformin (200 μM). Scale bar, 100μm. ** *p* < 0.01, *** *p* < 0.001. (**C**) The effects of metformin (200 μM) on H1299 and MCF-7 cells proliferation in vitro at 48 h. Cells were seeded into 96-well plates and cell proliferation was examined by CCK-8 assays. (**D**) Functional analysis of identified proteins in Mass spectrometry. (**E**) The conditioned medium (CM) of H1299 and MCF-7 cells was collected and concentrated, and then extracellular Hsp90α (eHsp90α) was measured by Western blot. Co.St (Coomassie brilliant blue) was used as a control. (**F**) The quantified results of the Western blot. (**G**) The effects of metformin (200 μM) on the expression of Hsp90α measured by qRT-PCR. (H-I) H1299 cells with luciferase activity were orthotopically injected into nude mice (*n* = 7 or 8/group). Mice were treated once daily with saline or metformin (250 mg/kg) by oral gavage. (**H**) Representative bioluminescent (BLI) images were acquired. Heat-maps indicated the intensity of bioluminescence from low (blue) to high (red). (**I**) Plasma was collected and extracellular Hsp90α was measured by ELISA assay. (**J**) MCF-7 cells were injected into nude mice (*n* = 8/group) and extracellular Hsp90α in plasma was measured by ELISA assay.

**Figure 2 cells-09-00144-f002:**
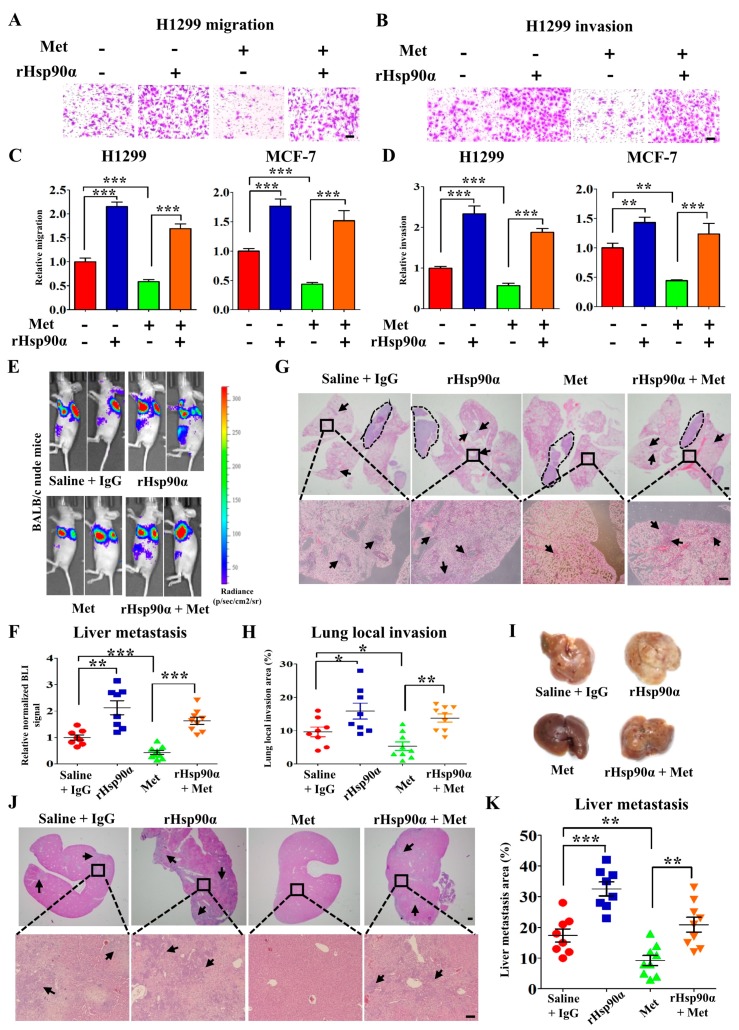
Recombinant Hsp90α reversed the inhibitory effects of metformin on tumor metastasis. Representative images of cell migration (**A**) and invasion (**B**) in H1299 cells treated with or without metformin (200 μM) and recombinant Hsp90α (10 ng/mL). Quantified results of cell migration (**C**) and invasion (**D**) in H1299 cells treated with or without metformin and recombinant Hsp90α. Scale bar, 100 μm. ** *p* < 0.01, *** *p* < 0.001. (**E**–**K**) H1299 cells with luciferase activity were orthotopically injected into nude mice (*n* = 8 or 9/group). Mice were treated with or without metformin (200 μM) and recombinant Hsp90α (10 mg/kg). (**E**) Representative bioluminescent images (BLI) and (**F**) quantified results of fluorescence intensity for liver metastasis. (**G**) Representative H&E images of lung local invasion, and the area in the circle is the primary tumor. (**H**) Quantified results of lung local invasion areas. Scale bar, 2 mm (above), 200 μm (below). (**I**) Representative images of liver tissue. (**J**) Representative H&E images for liver metastasis. Scale bar, 2 mm (above), 200 μm (below). (**K**) Quantified results of liver metastasis areas.

**Figure 3 cells-09-00144-f003:**
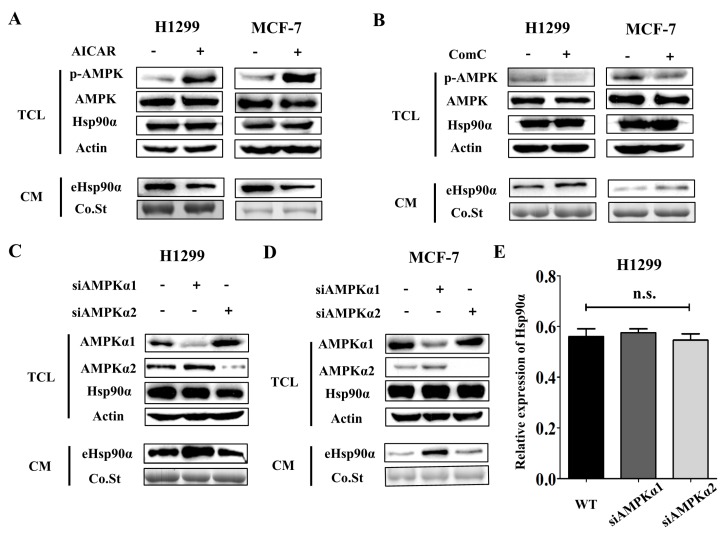
AMPKα1 but not AMPKα2 mediates Hsp90α secretion. (**A**) Extracellular Hsp90α was measured by Western blot in both H1299 and MCF-7 cells treated with or without AMPK activator, AICAR (200 μM). Western blots of AMPK, p-AMPK, actin, Hsp90α were also shown. p-AMPK meant the Thr172 phosphorylation, an indicator of AMPK kinase activity. TCL: total cell lysate, CM: conditioned medium, Co.St: coomassie staining. (**B**) Extracellular Hsp90α was measured by Western blot in both H1299 and MCF-7 cells treated with or without AMPK inhibitor, Compound C (2 μM). Western blots of AMPK, p-AMPK, actin, Hsp90α were also shown. (**C**–**D**) AMPKα1 and AMPKα2 were knocked down separately by siRNAs in tumor cells. Extracellular Hsp90α was measured by Western blot in wild type, AMPKα1-deficienct and AMPKα2-deficient H1299 cells (**C**) and MCF-7 cells (**D**). (**E**) The effects of AMPKα1 and AMPKα2 on proliferation were examined by CCK8 assay in H1299 cells. siAMPKα1 meant the siRNA for AMPKα1. siAMPKα2 meant the siRNA for AMPKα2.

**Figure 4 cells-09-00144-f004:**
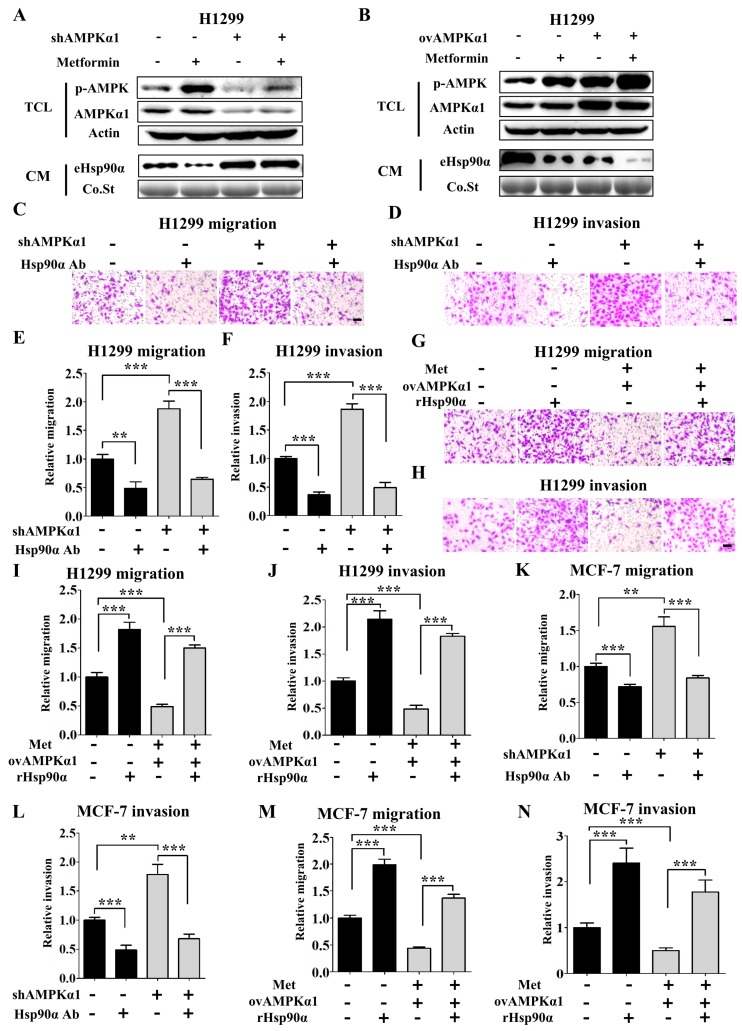
Metformin inhibits Hsp90α secretion in an AMPKα1 dependent manner. Extracellular Hsp90α was measured in the conditioned medium of AMPKα1 KD (**A**) and AMPKα1 OV (**B**) H1299 cells treated with or without metformin (200 μM). Western blots of AMPKα, p-AMPK and actin were also shown. Co.St (Coomassie brilliant blue) was used as a control. shAMPKα1 meant the stable knockdown of AMPKα1. ov AMPKα1 meant the stable overexpression of AMPKα1. Representative images of H1299 KD cells migration (**C**) and invasion (**D**) treated with or without Hsp90α antibody. Quantified results of H1299 cells migration (**E**) and invasion (**F**). Representative images of H1299 OV cells migration (**G**) and invasion (**H**) treated with or without recombinant Hsp90α. Quantified results of H1299 cells migration (**I**) and invasion (**J**). Scale bar, 100 µm. ** *p* < 0.01, *** *p* < 0.001. Quantified results of MCF-7 KD cells migration (**K**) and invasion (**L**) treated with or without Hsp90α antibody. Quantified results of MCF-7 OV cells migration (**M**) and invasion (**N**) treated with or without recombinant Hsp90α.

**Figure 5 cells-09-00144-f005:**
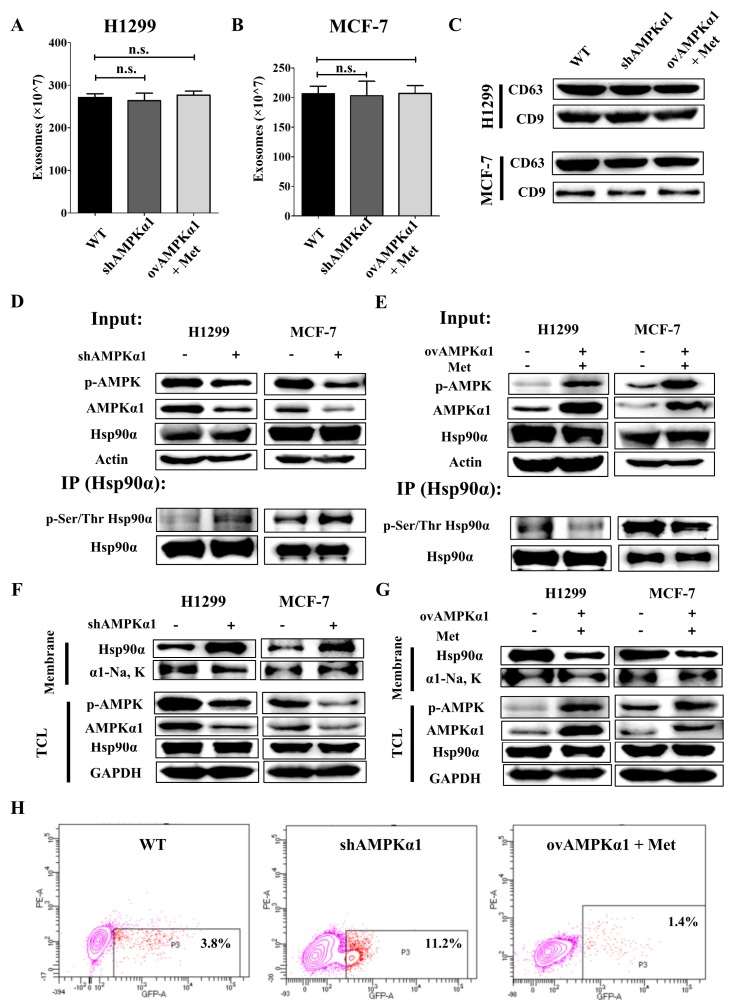
AMPKα1 decreases the phosphorylation level of Hsp90α and suppresses Hsp90α membrane translocation. The Quantified results of exosomes in H1299 cells (**A**) and MCF-7 cells (**B**) measured by Exosomes Quantified Kit. (**C**) Western blots of CD63 and CD9 in H1299 and MCF-7 cells. (**D**) Hsp90α was pulled down in H1299-WT, H1299-KD, MCF-7-WT and MCF-7-KD cells and the phosphorylation level at Ser/Thr was measured by western blot. (**E**) Hsp90α was pulled down in H1299-WT, H1299-OV, MCF-7-WT and MCF-7-OV cells treated with or without metformin and the phosphorylation level at Ser/Thr was measured by western blot. (**F**) Plasma membrane extractions of H1299-WT, H1299-KD, MCF-7-WT and MCF-7-KD cells were analyzed by western blot. Na, K-ATPase α1 was the plasma membrane marker. TCL: total cell lysate. (**G**) Plasma membrane extractions of H1299-WT, H1299-OV, MCF-7-WT and MCF-7-OV treated with or without metformin were analyzed by western blot. Na, K-ATPase α1 was the plasma membrane marker. (**H**) Hsp90α on the cell membrane was measured by flow cytometry in H1299-WT, KD and OV cells.

**Figure 6 cells-09-00144-f006:**
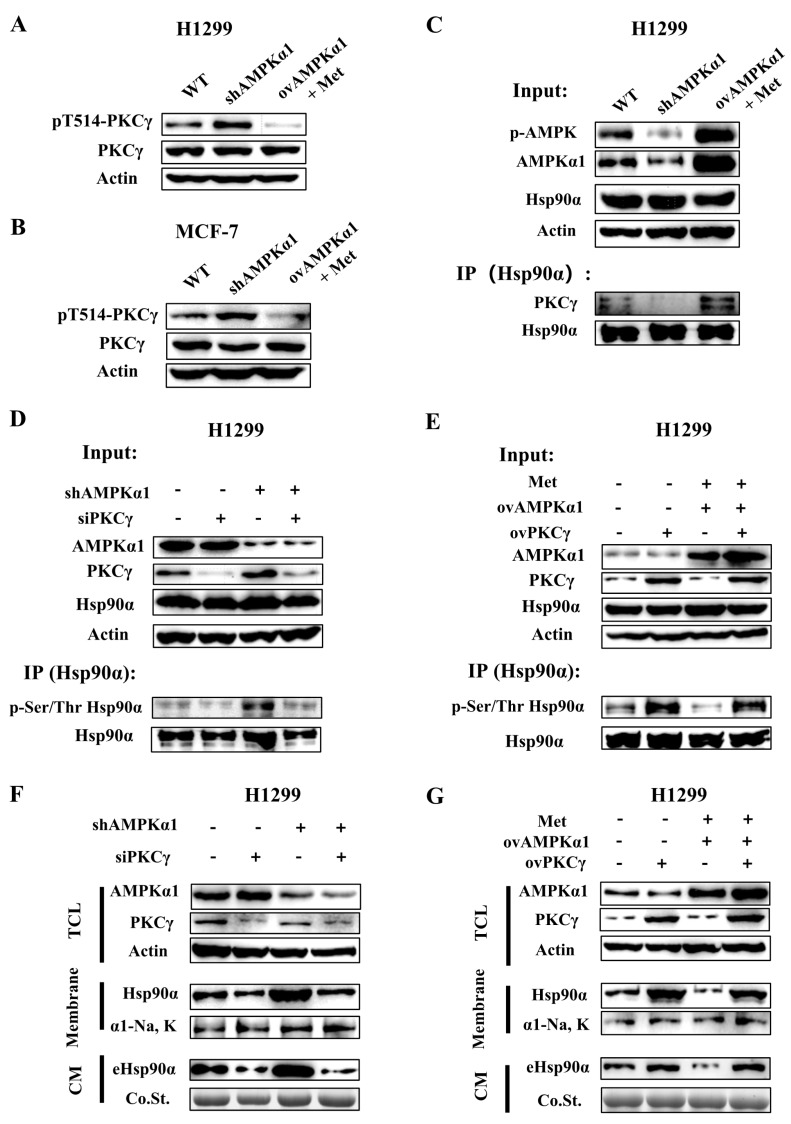
AMPKα1 inhibits Hsp90α phosphorylation, membrane translocation and secretion by suppressing the kinase activity of PKCγ. pT514-PKCγ, PKCγ and actin were measured by Western blot in H1299-WT, KD and OV cells (**A**) and MCF-7-WT, KD and OV cells (**B**) The level of pT514 is an indicator of PKCγ activation. (**C**) Hsp90α was pulled down in H1299-WT, KD and OV cells treated with or without metformin. PKCγ was measured by western blot. (**D**) Hsp90α was pulled down in WT, AMPKα1-KD, PKCγ-KD and AMPKα1-PKCγ-double KD H1299 cells. The phosphorylation level of Hsp90α at Ser/Thr was measured by western blot. (**E**) Hsp90α was pulled down in WT, AMPKα1-OV, PKCγ-OV and AMPKα1-PKCγ-double OV H1299 cells treated with or without metformin. The phosphorylation level of Hsp90α at Ser/Thr was measured by western blot. (**F**) Plasma membrane extractions and conditioned medium of WT, AMPKα1-KD, PKCγ-KD and AMPKα1-PKCγ-double KD H1299 cells were analyzed by western blot. Na, K-ATPase α1 was the plasma membrane marker. (**G**) Plasma membrane extractions and conditioned medium of WT, AMPKα1-OV, PKCγ-OV and AMPKα1-PKCγ-double OV H1299 cells treated with or without metformin were analyzed by western blot. Na, K-ATPase α1 was the plasma membrane marker.

**Figure 7 cells-09-00144-f007:**
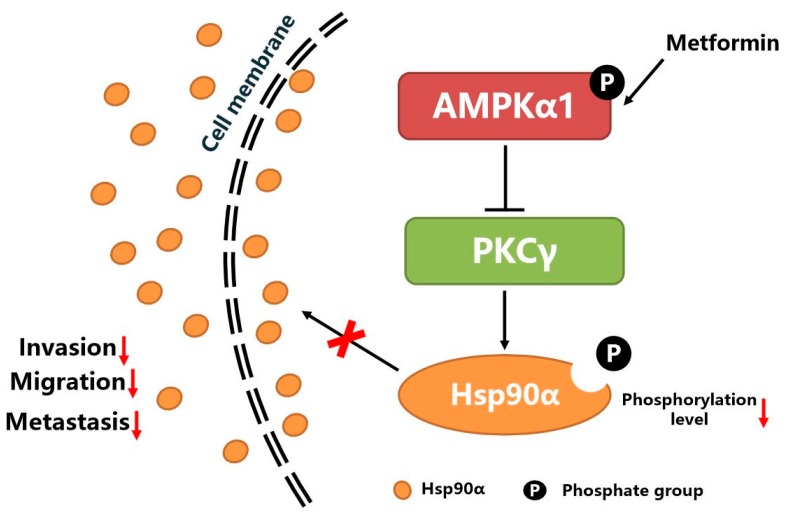
The graphical conclusion of the research: metformin inhibits tumor metastasis through suppressing Hsp90α secretion in an AMPKα1-PKCγ dependent manner.

**Table 1 cells-09-00144-t001:** Relative protein concentration in conditioned medium changed more than twofold after metformin treatment (<2.0) measured by MS.

Protein Name	Score		Coverage		Peptides		Change	Functions
	Ctrl	Met	Ctrl	Met	Ctrl	Met	Ctrl/Met	
THBS1	223.84	161.90	15.04	12.65	16	13	3.52	Promoting platelet aggregation, angiogenesis and tumorigenesis
Hsp90α	875.14	908.54	39.07	38.25	43	39	3.24	Promoting tumor cells migration, invasion and metastasis
CYR61	158.90	128.75	33.07	32.81	12	11	2.33	Extracellular matrix-associated protein, regulating cell adhesion, migration and proliferation
TIMP1	612.70	556.36	62.32	62.32	10	10	2.28	An inhibitor of the matrix metalloproteinases, promoting cell proliferation and wound healing
14-3-3 protein zeta/delta	353.79	332.20	72.65	72.65	23	23	2.11	Promoting lung tumor and breast tumor metastasis
LOXL2	120.47	100.85	21.83	17.44	15	11	2.08	Formation of crosslinks in collagens and elastin, promoting tumor metastasis and lymphatic metastasis

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
