# Peer review of "Metformin Inhibits Tumor Metastasis through Suppressing Hsp90α Secretion in an AMPKα1-PKCγ Dependent Manner"

_cells, 2020, doi:10.3390/cells9010144_

Round 1

Reviewer 1 Report

Dr. Gong et al. argue in this manuscript that metformin-induced activation of AMPKa inhibits phosphorylation of PKCg (threonine 514), which downregulates phosphorylation of hsp90a at Ser/Thr residues, leading to inhibition of Hsp90a’ secretion toward extra cellar space. As the secreted Hsp90a plays a role in migration and invasion of cancer cells, the story seems to explain the underlying mechanism behind the metformin-induced suppression of metastasis of certain tumors. The overall story is interesting and acceptable, the experiments were done carefully and extensively. Also, the discussion is well-balanced.

The only concern about this study is how the authors identified the hsp90 is Hsp90a but not Hsp90b. The information of recombinant Hsp90a protein and antibody to Hsp90a (D10) from Protgen company has some difficulty to obtain by PC search. Does the antibody recognize amino acid sequences specific to Hsp90a but not to Hsp90b? If so, which amino acid sequence? Also, was the same antibody used for immunoprecipitation of Hsp90? If identification of Hsp90a was done by Mass-Spec analysis, which fragment(s) specific to Hsp90a was identified?

The description Phosho-Ser/The-Hsp90 in Table S3 should be corrected because the antibody is directed to phospho - (Ser/Thr) residues of any proteins but not specific to Hsp90.

Reviewer 2 Report

Review of an article Metformin inhibits tumor metastasis through suppressing Hsp90α  secretion in an AMPKα1-PKCγ dependent manner

Manuscript investigates the role of metformin in tumor growth and suppression of invasion based on a numerous epidemiological studies. Manuscript address a novel mechanism of a metformin exerts it´s tumor metastasis inhibition by mediating tumor secretome analyzed here with a MS.

Authors revealed that metformin mediates tumor secretome via activating AMPK1 that in turn suppress PKC kinase activity towards HSP90 and more specifically suppress HSP90α secretion analyzed by mass spectrometry in vivo and in vitro.

This metformin activated AMPK mediated secretome is a novel finding that has been further characterized in this work by investigating the role of AMPKα1 kinase activity that has been shown here to inhibit the kinase activity of PKCg. AMPKα1 specifically is shown here to inhibit PKCg kinase activity towards HSP90α but not the isoform AMPKα2. Thus, the finding is not dependent on the genetic background of the tumor tissues. Hypophosphorylation of HSP90α by PKCg suppress the membrane association and further inhibits secretion or exocytosis of HSP90 of the cancer cell models H1299, human breast cancer cell lines MCF-7, but not in a  A549MDA-MB-231 models tested here.

The authors show the link between PKCg and HSP90 phosphorylation in the figure 6 by using PKCg siRNAs and HSP90 pulldowns.

Conclusion were made that that metformin inhibits tumor metastasis by suppressing Hsp90α secretion in an AMPKα1 dependent manner that is plausible by the results shown in the manuscript

Finding of AMPK activation mediated suppression of HSP90 secretion trough membrane localization is indeed a novel finding.

Methodology:

Methodology for the manuscript is plentiful and are carefully chosen to cover up the data needed.

Comment for the methodology: For a reason that is not explained on the manuscript the metformin concentration used here is quite high 200 mg/kg per day. Did the authors made any experiments with lower concentrations of metformin as there might be difficulties to translate the results for the human experiments. Also, it has been shown that some cancer cell lines don´t have a transport that is needed to translocate the metformin to the cell cytosol that might explain the need of high concentrations used here that might not be achievable by clinical settings.      

Comments:

Page 1. Phrase: Metformin can cause demethylation of DNA and lead to up-regulation of some encoding genes and non-coding RNAs [6]. If there are few upregulated genes, would be helpful for the reader that those genes are listed here.

Page 1. Phrase: Several studies also show that metformin can inhibit the epithelial to mesenchymal 38 transition (EMT) process [10]. If there are several studies, why only one reference which describes TGFβ regulation by metformin.

Figure 1. H. The representative bioluminescence images (BLI) were not quantitated. Is there a difference between BLI between Saline and Metformin group?

Table 1. Title text: “Identified proteins that changes more than twice before and after metformin treatment”. Should it be written: Relative protein concentration in conditioned medium changed more than twofold after metformin treatment (<2.0) measured by MS? Same comment fit on the phrase on the page 5. Line 202.  

Page 7. Line 248. Used word “eviscerated” in a number of places. Could it changed to “removed for”?

Figure 2. Title of the figure needs reformatting as in the results shown there is no measurement of HSP90α secretion but recombinant HSP90α has been administrated to the bloodstream of mice’s with or without a metformin.

Second comment for Figure 2: Why the BLI images are not quantitated in picture E?

Figure 6. Major finding that show the link between AMPK activation and PKCg. I would like to see statistical significance of the quantitated pulldowns of p-Ser/Thr HSP90α for the figure D. and Figure E. to verify that the major finding is statistically significant and representative images for the western are shown.

Reviewer 3 Report

This is a well-designed and well-written original study in which the authors showed that the anti-tumor effect of metformin can be associated with inhibition of H1299 and MCF-7 cell migration by suppressing Hsp90α secretion. Some remarks should be considered before publishing this paper.

1. The abstract should indicate specific cell lines for which metformin-induced migration inhibition has/ not been demonstrated.

2. The authors should explain why they decided to choose for study A549, H1299, MDA-MB-231 and MCF-7 cell lines.

3. Please try to identify the mechanism by which metformin was not able to inhibit migration of  A549 and MDA-MB-231 cells.

4.In Figure 3 the following abbreviations are used: TCL, CM, Co.St. Please explain their meaning. 

5. Why  did you change the reference protein in experiment that results are depicted in Figure 5F (GAPDH instead of actin). Again, there are unexplained abbreviations in Figure 5F.

6.In order to sum up your results, please include to your paper a figure representing signaling pathway, including crucial molecules identified in this study that is targeted by metformin.
